# Do chronic illnesses and poverty go hand in hand?

**Ruwan Jayathilaka**⊕*, **Sheron Joachim, Venuri Mallikarachchi, Nishali Perera**⊕**, Dhanushika Ranawaka**

Department of Business Management, SLIIT Business School, Sri Lanka Institute of Information Technology, Malabe, Sri Lanka

* ruwan.j@sliit.lk

**Data Availability Statement:** This data set is not publically available to download. However, interested parties can obtain this data file after visiting the Department of Census and Statistics in Sri Lanka or following the Microdata Dissemination

## Abstract

In the global context, the health and quality of life of people are adversely affected by either one or more types of chronic diseases. The chronic pain associated with diagnosed patients may include heavy medical expenditure along with the physical and mental suffering they undergo. Usually, unbearable amounts of medical expenses are incurred, to improve or sustain the health condition of the patient. Consequently, the heavy financial burden tends to push households from a comfortable or secure life, or even from bad to worse, towards the probability of becoming poor. Hence, this study is conducted to identify the impact chronic illnesses have on poverty using data from a national survey referred as the Household Income and Expenditure Survey (HIES), with data gathered by the Department of Census and Statistics (DCS) of Sri Lanka in 2016. As such, this study is the first of its kind in Sri Lanka, declaring the originality of the study based on data collected from the local arena. Accordingly, the study discovered that married females who do not engage in any type of economic activity, in the age category of 40–65, having an educational level of tertiary level or below and living in the urban sector have a higher likelihood of suffering from chronic diseases. Moreover, it was inferred that, if a person is deprived from access to basic education in the level of education, lives in the rural or estate sector, or suffers from a brain disease, cancer, heart disease or kidney disease, he is highly likely to be poor. Some insights concluded from this Sri Lankan case study can also be applied in the context of other developing countries, to minimise chronic illnesses and thereby the probability of falling into poverty.

## Introduction

Chronic diseases can be broadly defined as conditions that last for a year or more and require on-going medical attention that limits activities of daily living [1]. Poverty is the state of one who lacks a usual or socially acceptable amount of money of material possessions [2]. Chronic diseases can be perceived as a global predicament, where two thirds of deaths that occur in the world today are because of one or more types of chronic diseases. However, the harmful effect of chronic illnesses is not limited to premature death; it is also responsible for other confrontational effects associated with the economic well-being of an individual, households and the society as a whole. The most devastating outcome that people affected with chronic illnesses

policy in DCS. Interesting parties should fill the D. R.A Form which is available at the following link. https://www.ilo.org/surveyLib/index.php/catalog/1076/download/7295 This study used the Household Income and Expenditure Survey – 2016 Interesting parties should be completed the D.R.A form electronically(except for signature) then after filling the form, It should be printed, signed, scanned and sent to: [Director General, Department of Census and Statistics, No 109, Rotunda, Tower, 5th Floor, Galle Road, Colombo 04, Sri Lanka.] Or E-mail scanned copy to: dgcensus@statistics.gov.lk After requesting for the survey data from Department of Census and Statistics in Sri Lanka, they will evaluate the application before providing the access to the data. Moreover, authors did not receive any special privilege of obtaining these data files.

**Funding:** The authors received no specific funding for this work.

**Competing interests:** The authors have declared that no competing interests exist.

experience is the inability to perform daily tasks, socialise freely as they used to be and loss of independence.

The burden of being chronically ill does not only lie upon and remain within the ill person but his whole family as well; therefore, the entire household ends up becoming indirect victims. This is due to the extensiveness and long duration of chronic conditions which require continuous caregiving to the patient and out-of-pocket expenses for medication. On the one hand, ill health adversely affects productivity of employees which results in lower performance and decrease in pay and disposable income they receive. On the other hand, companies as employers will always prefer to recruit employees capable of contributing with their maximum potential in order to reach corporate goals. According to this logic, a chronic condition can arouse the risk of pushing households towards poverty.

People suffering from these diseases end up making choices that are challenging or rather contradictory; either they have to ignore the condition they are in, avoid medical treatment, and face premature death by investing their earnings in satisfying their key needs and wants, or to seek health care treatment by processing with out-of-pocket expenditures, thus, drag their families to the ill effects of poverty. This burden is severe when people suffer from multiple chronic conditions and disabilities which require supplementary health care services and attention frequently, to prevent the condition from becoming critical.

## Problem statement

The issues associated with chronic illnesses are projected to rise particularly fast in the coming years, especially in developing countries like Sri Lanka, creating significant barriers to growth and development. According to the World Health Organisation [World Health Organization [3]], four out of five chronic disease deaths that occur in the world today are from low and middle-income countries (LMICs) like Sri Lanka. The low-income households are at risk the most in developing chronic diseases and for premature deaths. Such households are more vulnerable for several reasons, including their inability to cover medical expenses and diminished access to healthcare facilities.

Currently, the extent to which chronic illnesses would affect the level of poverty in Sri Lanka is a less attention driven subject and an unexplored area by the past researchers. Thus, this study focusses to contribute to the above-mentioned empirical gap by examining the growing toll of chronic diseases and its relevance to poverty, with specific attention to Sri Lanka.

## Objective

The objective of this study is to investigate the impact chronic illnesses on the level of poverty. As such, this research differs from existing studies to date, and contributes to literature in four ways. Firstly, chronic illnesses being a severe health condition that persists for a period of one year or more, require households to incur continuous caregiving and medical treatments on behalf of their patients. Such treatments are mandatory as these patients can become severely ill and helpless for a considerable period of time. Consequently, this setting creates numerous barriers to perform routine activities on a daily basis for both parties, i.e. for the sick person and his family members. At present, an individual is prone to be affected by more than one chronic condition with conditions worsening [4]. Therefore, non-communicable diseases (NCDs) have become a major health issue in the 21st century which requires the attention of regulatory bodies of a nation such as the government, healthcare sector and other policy makers. This is noticeable especially in LMICs such as Sri Lanka, due to the relatively high exposure to risks and limited access to better healthcare facilities for the general public.

Secondly, no prior research study has been conducted to date with regard to the area under consideration, addressing the local arena. This study will be the first attempt of this kind carried out with data at a broad level, covering the entire country, according to the information available to researchers.

Thirdly, according to the health goal 'SDG 3' in line with the Sustainable Development Goal (SDG) profile of Sri Lanka issued stipulated by the WHO South East Asian Region in 2017, currently the likelihood to die from NCDs before the age of 70 is 17.7%. This number is expected to rise in the coming years [5]. Hence, at the completion of the study, the findings of this research will provide valuable insights to the Government of Sri Lanka (GOSL) for introduction, planning, implementation and monitoring of new policies related to healthcare. It can also assist to eliminate or curb the probability of occurrence of NCDs among the general public by spreading awareness among the society.

Finally, the findings will be helpful, particularly to the healthcare sector and policy makers who endeavour to recuperate economic setbacks and quality of life in the aftermath of the coronavirus (COVID-19) global pandemic, in early 2020.

The remaining sections of this paper are organised as follows. Section 2 describes the literature review and underlying theories while emphasising on significance of this study, while Section 3 presents data and methodology. Section 4 assesses the empirical results and the discussion, whereas Section 5 presents the concluding remarks with policy implications.

## Literature review

An element of the literature search strategy of this study focussed on the initially identified 159 publications through an extensive and a thorough literature hunt; a keyword search consisted of eight dissertations and five conference papers. This literature search was conducted in a number of reputed journal databases such as, Science Direct, Emerald Insight, Business Insight, JSTOR, SAGE Premier, Research Gate, Medline etc. The search terms used were: (chronic illnesses OR non-communicable diseases OR illnesses OR severe diseases OR poverty OR chronic disease induced poverty OR low- and middle-income countries) AND (probit OR regression OR binary probit model).

Thereafter, a screening process was executed in order to strain out and thereby, shortlist the most relevant publications. Accordingly, 135 publications were identified as either related to types of chronic diseases or chronic disease induced poverty; the rest of the 24 publications were excluded as these failed to fulfil the inclusion criteria. Fig 1 shows the flow diagram which included the number of studies that were located, retained, and discarded at each stage of the literature review. Four overlapping publications, six publications with insufficient information, whereas nine publications that discussed about other NCDs and five irrelevant publications were excluded.

The remaining 135 publications underwent another screening process, where 113 publications were identified based on the relevance to the title and abstract. After excluding 29 publications, the remaining 106 were re-screened in order to identify the eligible literature based on key words and the context. At this level of screening, 65 publications were recognised, as overall, which met these criteria; the recount was filtered to 53 publications as the rest were not contextually relevant. Accordingly, the final residual 53 papers form the publications used in developing the literature review of this study.

Chronic illnesses such as diabetes mellitus, cardiovascular disease, cerebrovascular disease, cancers, chronic respiratory diseases, mental illnesses, and other NCDs have become the major cause of death and disability of citizens in countries worldwide [6], where more than two third of all deaths are caused by a certain type of a chronic disease [7]. More than 41

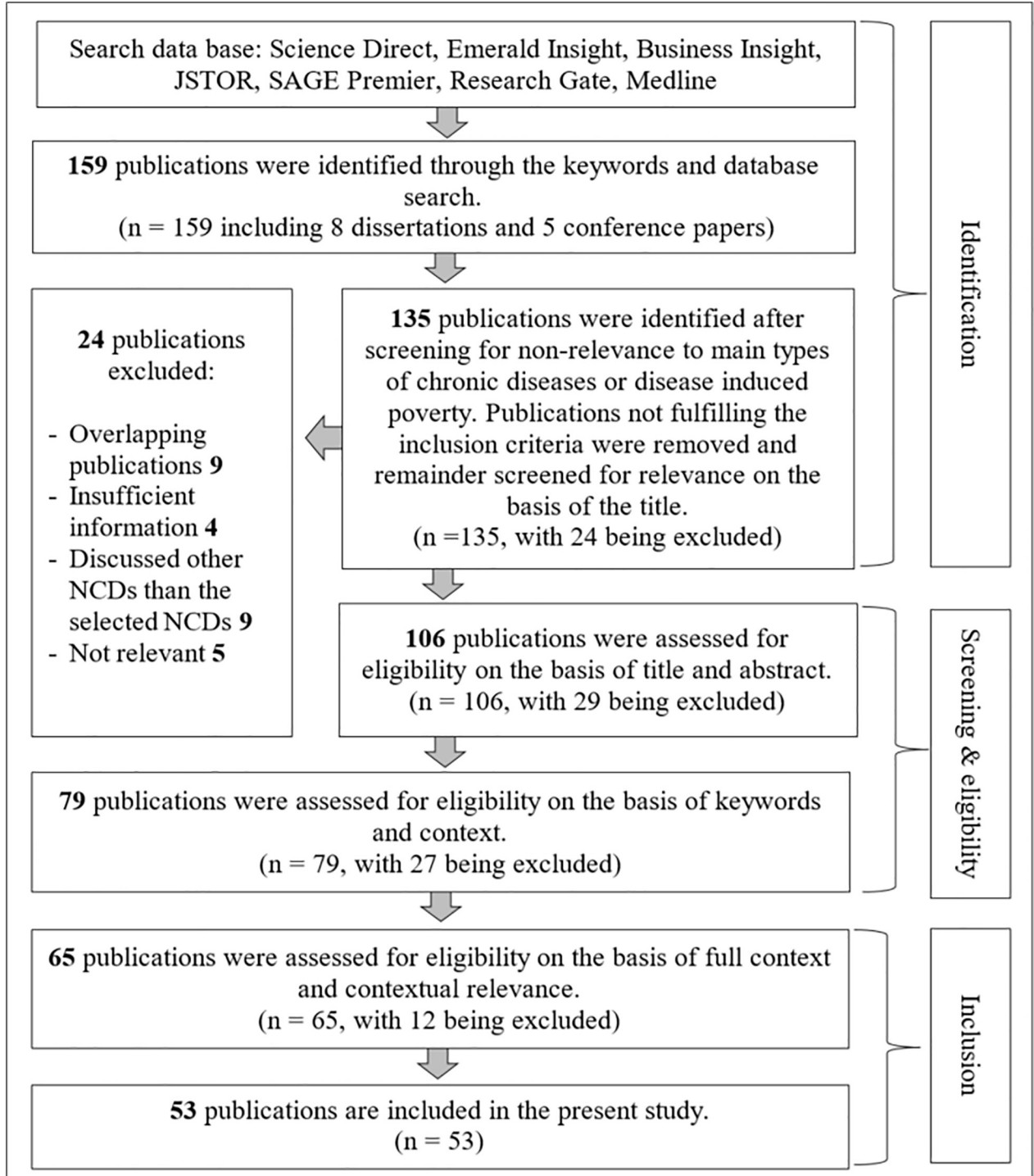

**Fig 1. Literature search flow diagram.** Source: Based on authors' observations.

million people have died from NCDs in 2016, out of which 15 million of these deaths have occurred between the age of 30–70 years, mainly from cardiovascular diseases (31%), cancers (16%), chronic respiratory diseases (7%), diabetes (3%) and other NCDs (15%) [3]. According to WHO estimates, nearly 78% of total deaths caused by chronic diseases have occurred in

LMICs in 2016 [8]. This is due to mainly four types of NCDs–cardiovascular diseases, cancers, chronic respiratory diseases and diabetes. On average, NCDs are projected to be the major cause of disease burden worldwide, exceeding communicable diseases, puerperal, prenatal, and food diseases in every country; chronic diseases are recognised as the origin of disability as 68% of people living with disability worldwide, and 84% of people living in LMICs [9]. If current growth trends persist by 2020, 7 out of every 10 deaths in developing countries will be attributed to NCDs and NCDs deaths worldwide by 2030 will be 52 million [10]. A research study in the USA states that, 12% of the child population is likely to suffer from a certain type of a NCD, with asthma being the most common illness next to allergy and obesity; this is predicted to rise in the coming years [11]. Thus, the increasing prevalence of multiple chronic conditions will adversely affect every country across the globe and result in high healthcare utilisation and increasing costs.

The effect of the growth of different chronic illnesses and disorders befalls on the world at large. Especially, its impact on LMICs is possibly high due to various reasons. Confirming the above fact, a study claims that the increasing global burden of diseases caused by all forms of chronic conditions may impose a heavy financial burden on households, specifically in LMICs against developed nations [12]. It has been found that medical expenses represents a substantial proportion of economic costs pertaining to treating chronic illnesses in poor countries; hence, the growth of chronic illnesses will have an adverse impact on the lives of people living in such countries [13]. According to Burki, Khan [14], although most chronically ill patients are found among the non-poor, it cause more severe consequences on middle income and low-income earning people within the country. Thus, these result in increasing the probability as well as vulnerability of becoming poor. In addition, most of the LMICs have a considerably low level of public expenditure, inadequate health insurance and low coverage of health care services compared to well off nations. The insufficiency in public health services and expenditures has caused victims to experience high amounts of out-of-pocket expenditure, as they need to acquire private health care services which are typically unbearably expensive [15]. Mostly, these expenditure consists of medication costs for patients who are in need of regular and on-going treatments, payments for medical and allied health services, purchase of medical devices and other related services [16]. Hence, the expenditure on chronic diseases has impacted heavily on households, especially in LMICs due to excessive expenses compared to the meagre income they earn.

Certain studies claim that the literature reflects the influence and behaviour of various demographic characteristics on chronic illnesses. Previous empirical findings disclose of a rapid demographic shift towards an ageing population forecasted to grow, which will turn out to be a quarter of the population by the year 2050 [17]. Pati, Agrawal [18] emphasise that the average number of non-communicable chronic diseases increase with age. When provided with statistical data, 1.3% of patients are in the youngest group (18 to 29 years) and 30.6% in the oldest group (70 years and below). Therefore, an individual in the age category of 33–44, has a higher likelihood of spending on medications resulting in high out-of-pocket medical expenditure [19]. When taking gender into account, the studies state that the prevalence of chronic diseases is high among the females when compared with the chronically ill male population [9, 20]. Peek, Drum [21] assert that, among all chronically ill patients, nearly two third of the patients were females which is 66.1% to be precise, where the mean age was 58.3 years. As a result, elderly females are more vulnerable to diseases and likely to spend higher out-of-pocket health expenditure than males.

Marital status of households is another variable impacting on chronic conditions. When combining the marital status of households along with different types of chronic diseases, most researchers have discovered that those married are more likely to report having a

disability than those unmarried [9]. Another study reveals that being married decreases the possibility of having access to patient centred medical homes when compared against divorced and separated individuals, thus, increasing the chances of being a victim of chronic illnesses [22].

The education level of the households is another socio-economic factor that is associated with the occurrence of NCDs. A research conducted in Hong Kong proves the fact that, lesser the population has progressed with their basic education, the more likely they are to face the risk of chronic illnesses and have higher chances of ending up with multi-morbidity [23]. A research carried out in USA further elaborates that, lesser the number of years of education of a person, the more vulnerable he is, for facing a chronic disease [22]. Therefore, higher education attainment of the head of the household results in better income and living conditions, thus, declining the financial burden on members with chronic illnesses [24]. However, in contrast, another study held in Maine State in the USA, indicated its inability to figure out a correlation between the educational progress and prevalence of chronic illnesses; it argued that since the state already recorded an education level much higher than the national average and therefore, the prevalence has occurred from other factors [25]. However, most researchers suggest that household heads having beyond an average education have relatively, better and a stronger communication attitude towards health-related behaviours; thus, it makes the education level a much stronger protective factor in facing chronic illnesses. On the other hand, low education means individual may have to do jobs that involve strenuous labour or work in hazardous conditions to earn a living, which increases their chances of getting a chronic illness. Hence, households with a sound educational level have lesser severance of growth in chronic diseases, as health payment opportunities are available for them, despite been in either the rural or urban areas.

Furthermore, chronic illnesses and disabilities can require more commitment for the disabled family member and medical expenses, thereby result in adverse economic consequences to collapse the economic stability of a household—such as unemployment, change in the state of employment, reduction in employee payment, out-of-pocket medical expenses, home modification expenses, etc. A reduction in labour units and capital accumulation [26] caused due to less affordability for treatments and other health related setbacks would restrict the economy towards developing [27]. Thus, the impact of chronic illnesses towards the economic growth of a country is high. Additionally, another study affirms that, chronic illnesses caused around 40% of reduction in employment related activities; it further resulted in declining income from employment connected with the high medical expenses, which households need to incur on medicine and other treatments [9]. The finding also emphasises on the fact that, out-of-pocket medical expenditure is high among unemployed people or people who have no source income, than those who are employed. A research conducted in Australia highlights that, increasing prevalence of chronic diseases such as heart diseases, diabetes and mental health issues can have a negative impact on the labour force participation in developed nations [28]. Additionally, one of the recent studies conducted in the USA asserts that, treatment of chronic illnesses and productivity losses occur due to persistence of such chronic illnesses; this cost the US economy more than USD1 trillion dollars annually [29]. Therefore, as a consequence of NCDs, victims become partially or fully unemployed resulting their usual level of income to drop, which limits the ability in satisfying their basic needs and wants.

Nevertheless, income level of households is another socio-economic variable that can push them to poverty caused by chronic conditions. A research specifies that the prevalence of chronic diseases is high among the people who live in rural areas than those in urban areas, due to the existence of income inequalities. It also highlights that in Bangladesh, a vast majority of people who suffer from chronic illness and associated disabilities falls under the lowest two

wealth quintiles of the society [9]. This fact is further confirmed by empirical evidences of another research, which implies that low socio-economic status causes a high financial toll on the individuals [30]; hence, can result in diminished physical functioning and worsen the chronic burden when compared against individuals with high or middle income [31]. Another study claims that, having a low income also hinders adoption of healthy habits paving the way to poor health conditions [32]. Thus, it asserts that households with low income levels are at risk the most in facing a certain type of chronic disease. This is due to the heavy financial burden on household and high out-of-pocket medical expenditures.

Moreover, the heavier the burden, people are more likely to opt for quick financing, thus, borrow or sale of their assets can even cause a long term burden [12]. Such a setting drags people and their families affected into inflaming poverty, further trap them in, and not enable them to be relieved, but stick them to the so-called financial distress. Furthermore, high levels of financial stress, medical debt and bankruptcy can be found among people who do not own a proper health care insurance in developed countries such as, USA, Korea and Russia. Nevertheless, certain well-off economies like Australia has been able to compensate chronically ill people and their families from the financial burden they would face, by their publicly funded health care and social security arrangement [16]. Thus, the chronic burden could be curbed in LMICs, if such countries establish proper healthcare management systems.

In many cases, households with chronically ill members have a higher possibility of encountering health expenditure related poverty than families whose members are healthy. This situation is consistent in both the developed and developing countries [33], especially seen in countries like China [34, 35]. The situation in rural China is more critical as the economic burden due to NCDs has reached its peak by letting the middle-income earning families to fall into poverty, and families already in poverty to sink even deeper [36]. Likewise, with the substantial percentage growth of chronic patients in households, rising medical expenditure create poverty among such households [37]. Henceforth, since expenses are a drain on income they earn, chronic conditions not only impose a heavy economic burden on the diagnosed patients but much more on their families and society, especially in LMICs.

Overall, expenditure on chronic illnesses and lack of health care services have caused an unbearable burden on households in LMICs. In preventing and treating chronic illnesses, few barriers those would plug the path of uplifting the lives of the affected. Among the many factors, excessive prices and unavailability of medicines can be described as the key barriers to gain access in order to treat patients suffering from NCDs, especially in many LMICs. Furthermore, Australia being a well-developed nation still struggles with the issue of high medication costs, where low income earning households without entitlement for concessions spend 5% to 26% of their total discretionary income on medicines [38]. Lan, Zhou [37] assert that, low income earners with limited income and living in rural areas have a higher tendency to be victims of health payment induced poverty rather than income earners living in urban areas, since they are closer to the poverty line. The findings of another research depict that, households are more likely to spend 11% of their total household budget on healthcare and medications, whereas 50% of the occupants tend to spend 7% of their monthly per capita consumption expenditure on different illnesses [39]. Fong [40] describes that out of all NCDs, diabetes, cancer, cardiovascular disease and hypertension account for majority of the out-of-pocket medical spending especially among the elderly population. Also, flaws in health financing systems, unreliable and undependable medicines supply schemes as well as poor prescribing practices can be recognised as obstructions in accessing medications [41]. Moreover, Abegunde, Mathers [26] affirm that even if necessary information is available in free markets, patients lack knowledge in negotiating with health officers, doctors and other professionals

regarding the relevant treatments. Henceforth, having pragmatic yet firm healthcare policies is vital to any nation for alleviation of chronic illnesses and health payment induced poverty.

In a study based in Sri Lanka, Pallegedara [42] examined the effects of chronic NCDs on out-of-pocket health expenditures of households. Findings revealed that medical poverty is high among chronic NCDs. In another study, Pallegedara and Grimm [43] stressed that older persons are more inclined to suffer from chronic diseases. The two-part model based on the 2012/2013 household survey was carried out to examine the association of NCD-prevalence and healthcare utilization with household consumption [44]. They found that the relationship between private healthcare utilization and household consumption was negative. In another study, Kumara and Samaratunge [45] investigated the patterns and determinants of the burden of expenses in households. Findings confirmed that the burden of expenses does not vary significantly with the variation in income.

Though numerous evidences indicate a rapid growth in chronic diseases, literature is limited to the extent how households experience financial burden arising due to chronic diseases and disabilities. Thus, this study will focus to contribute to this empirical gap by examining how different types of chronic illnesses can impact on poverty, in the Sri Lankan context. Table 1 represents some of the variables related to past research studies on chronic illness or poverty.

Although extensive research has been conducted on chronic illness and poverty, most literature reviewed above has focussed on chronic illness or poverty distinctively and not on a combined perspective. Therefore, there is a need to contribute with new findings to this literature gap in the Sri Lankan context.

## Data and methodology

The study aims of measuring the impact of different types of chronic illnesses towards the level of poverty in Sri Lanka based on quantitative data gathered from the latest HIES; this is the ninth in the series, conducted in the year 2016 by the DCS under the strict guidance of National Household Survey Programme (NHSP) in Sri Lanka. The sample consists of 25,640

**Table 1. Summary of literature: Variables and supporting research articles.**

| Variable | Past research studies |
|---|---|
| 1. Age | Bleich, Koehlmoos [17], Pati, Agrawal [18], Liu, Rao [36], Sultana, Mahumud [9], Malon, Shah [25], Wang, Sun [41], Chung, Mercer [23], Almalki, Karami [22], WHO [46], Habibov [47], Jayasinghe, Selvanathan [48] |
| 2. Gender | Sultana, Mahumud [9], Abegunde, Mathers [26], Malon, Shah [25], Lan, Zhou [37], WHO [46], Font and Gil [49], Peek, Drum [50], Jayasinghe [51] |
| 3. Marital status | Sultana, Mahumud [9], Wang, Sun [41], Lan, Zhou [37], Almalki, Karami [22], Jayathilaka, Selvanathan [52] |
| 4. Employment status | Sultana, Mahumud [9], Chung, Mercer [23], Malon, Shah [25], Lan, Zhou [37], Zhang, Zhao [28] |
| 5. Income level | Sultana, Mahumud [9], Kemp, Preen [38], Almalki, Karami [22], Chung, Mercer [23], Malon, Shah [25], Wang, Sun [41], Christopher, Himmelstein [53], Fong [40], Perruccio, Katz [54], Kahn, Vest [55], Kim and Richardson [31], Kumara and Samaratunge [44] |
| 6. Educational level | Chung, Mercer [23], Malon, Shah [25], Lan, Zhou [37], Almalki, Karami [22], Parodi, Parodi [24] |
| 7. Ethnicity and Religion | Murphy, Mahal [12], Abegunde, Mathers [26], Bloom, Chen [27], Arrey, Bilsen [56], Bailey, Doyle [57], Coats, Downey [58], Druedahl, Yaqub [59], Nguyen, Paul [60], Shavers, Bakos [61] |
| 8. Lifestyle | Kankeu, Saksena [62], Bleich, Koehlmoos [17], Raghupathi and Raghupathi [29], WHO [46], Swindle, Shapley [63] |

Source: Authors' compilation.

housing units which includes 21,756 responded households, covering all 25 districts in the country and aims to analyse the seasonal and regional differences of income and expenditure levels together with buying patterns of households. This survey was conducted between the period from January to December in 2016.

The survey questionnaire mainly concentrates on three major criteria; demographic characteristics, household expenditure spent on food and non-food, and the household income earned in monetary and non-monetary terms. Two stage stratified sampling was employed in the sample design segregating the population into different stratas based on several characteristics such as age, income, geography etc. Here, the main area for stratification is on district basis whereas, urban, rural and estate sectors in each district are the selection domains [64].

The researchers employed probit model which was first introduced by Chester Ittner Bliss in 1935 and later used in many studies according to the past literature in achieving similar research objectives [52]. Hence, it is reasonable to assume that probit model is effective in this study and the validity of findings are reliable. Probit regression is a way of performing regression for binary outcome variables that are commonly identified as dependent variables, having two possibilities such as poor or non-poor. This model sets a parameter based on the change that occur within the binary variable, as a result of a change in independent variables [65].

$$P_i = \frac{1}{\sqrt{2\pi}} \int_{-\infty}^{z_i} e^{-s^2/2dt} \tag{1}$$

where:

$P_i$ = the probability that the dummy variable $D_i = 1$

$z_i = -\emptyset^{-1}(P_i) = \beta_0 + \beta_1 X_1 + \beta_2 X_2 + \cdots + \beta_n X_{ni}$

s = a standardised normal variable

The Probit model was developed with a dummy variable where poor is denoted as 1 and non-poor as 0, where people who live below the national poverty line, i.e. LKRs 4,166 per month are considered as poor. In empirical specification of this study, the decision of which variables to include is based on exploratory analysis. Table 2 shows the possible explanatory variables which are expected to have an effect on household poverty in the context of Sri Lanka, including socio-demographic, socio-economic, location and type of chronic illness variables. The forward step-wise regression technique selects the significant variables. New variables for selection with p-value <0.10 and previously selected variables for removal with p-value ≥0.15. The goodness-of-fit of the models is evaluated using an overall goodness-of-fit statistic developed by Ben-Akiva, Lerman [66] and the model with the highest goodness-of-fit value will be selected for this analysis.

## Results and discussion

Results are estimated based on probit model to achieve the prime research objective, i.e. the impact of various chronic illnesses towards the occurrence of poverty in Sri Lanka. Statistics of the DCS testimony that, at present, the Official Poverty Line (OPL) stands at SLRs.4,166 per month per individual. In other words, if the real per capita expenditure of a person living in a household falls below the value of the OPL or SLRs.4,166, he is considered to be poor in Sri Lanka. Estimates of the HIES disclose that, 843,913 individuals living in 169,392 housing units claim to live in poverty in 2016 and overall, represent 4.4% of the total household units [64]. Table 3 describes the population of households living in poverty in Sri Lanka in each sector as a percentage, for the year 2016.

Therefore, data used for the result estimation consists of 957 poor households and 20,799 non-poor households. Table 3 depicts that the highest percentage of country's population who

**Table 2. Variable definitions for household dataset.**

| Variables* | Description | Expected signs |
|---|---|---|
| *Socio-economic and demographic characteristics* | | |
| Chro_ill_patients | 1 if a chronically-ill patient; 0 if not. | (-/+) |
| Males_HH | Proportion of males in the household. | (-/+) |
| Elders_HH | Proportion of household members those who are above 66 years. | (+) |
| Pr_hh_working | Proportion of household members working; (employed household members/Labour force in the household). | (-) |
| Male_headed | 1 if a male headed household; 0 if not. | (+) |
| Head_age | Age of the household head (in years). | (-) |
| Maritalstatus_HH | Marital status of the household- 1 if married; 0 if not. | (-) |
| Edu_level | Household head's education level (years). | (-) |
| Ethnicity | Separate dummy variables for Sinhala, Sri Lankan Tamil, Indian Tamil, Sri Lankan Moors, Malay, Burgher, and Other; Other is the reference category. | (+) |
| Religion | Separate dummy variables for Buddhist, Hindu, Islam, Roman Catholic/Other Christian, and Other; Other is the reference category. | (+) |
| Health_exp | Health expenditure as a proportion of total expenditure. | (-) |
| Head_chronic | 1 if household head is a chronic patient; 0 if not. | (-) |
| *Geographical location* | | |
| Sector | Separate dummy variables for Urban, Rural, and Estate sectors; Estate is the reference category | (+/-) |
| District | Separate dummy variables for Colombo, Gampaha, Kalutara, Kandy, Matale, Nuwara_Eliya, Galle, Matara, Hambantota, Jaffna, Mannar, Vavuniya, Kilinochchi, Batticaloa, Ampara, Trincomalee, Kurunegala, Puttalam, Anuradhapura, Polonnaruwa, Badulla, Moneragala, Ratnapura, and Kegalle districts; Mullaitivu is the reference category. | (+/-) |
| *Type of chronic illness* | | |
| High_sev_diseases | 1 if the patient suffers from either heart diseases, blood pressure, cancer, kidney diseases, diabetes or asthma; 0 otherwise. | (-/+) |
| Brain_diseases | 1 if the patient suffers from either epilepsy, mental retardation or severe headache. | (-/+) |
| Ent_diseases | 1 if the patient suffers from either eye diseases, ear diseases or catarrh. | (-/+) |
| Otherdiseases | 1 if the patient suffers from either arthritis, stomach diseases or hemorrhoids. | (-/+) |

Source: Authors' compilation.

Note

*HH-household head.

live in poverty is from the estate sector with 91.32%, followed by the rural sector and urban sector which are 4.72% and 1.60%, respectively. The basic characteristics of each variable are presented in S1 Appendix. When socio-economic and demographic characteristics of households in Sri Lanka are taken into account, on average 74.41% of the households are male dominant and the average age of the household head in a household is 53 years. Additionally, the highest number of households located in the rural sector is 79.95% followed by the urban sector which is 15.76%. In Sri Lanka, 34.11% of individuals living in households suffer from severe

**Table 3. Level of poverty among Sri Lankan households 2016.**

| Level of poverty | Monthly expenditure | Percentage | | | |
|---|---|---|---|---|---|
| | | Overall | Urban | Rural | Estate |
| Poor | <SLRs.4,166* | 4.40 | 1.60 | 4.72 | 91.32 |
| Non-poor | >SLRs.4,166 | 95.60 | 98.40 | 95.28 | 8.68 |

Source: Author's calculation based on the DCS [64].

Note

*Official poverty line of Sri Lanka.

chronic illnesses such as heart diseases, blood pressure, cancer, kidney diseases, diabetes or asthma when compared with various other chronic disease types.

The initial probit model was estimated using all the independent variables and results are shown in S2 Appendix. For the variable selection for the final probit model, forward stepwise technique was adopted with p-value <0.10 and previously selected variables for removal with p-value ≥0.15. Insignificant variables such as Burgher, Buddhist, Kilinochchi and Rural were excluded when arriving at the final probit model. The estimation results of the final probit model are presented in Table 4. The area under the Receiver Operating Characteristic curve (ROC) is found to be 0.7969, which can thus be inferred that the estimated final probit model fits aptly to explain the link between the different type of chronic illness and poverty.

Furthermore, as shown in Fig 2, the areas under the ROC can be identified as 0.7969. In other words, area under the ROC depicts a higher value which is 0.7969. Hence, it can be concluded that the estimated probit model for household units suits appropriately to describe the connection between different types of chronic illnesses and poverty.

According to estimates of coefficients given in Table 4, socio-economic and demographic factors such as, chronically-ill patients, proportion of males in the households, proportion of household members working, age of the household head, educational level of the household head, health expenditure as a proportion of total expenditure, and chronic patient household head have a negative effect on being non-poor. In contrast, variables such as proportion of household members those who are above 66 years, the male headed households, married head, and ethnicity and religion categories have a positive effect on being non-poor. The estimated marginal effect on being poor is 0.53 percentage points higher for male headed households when compared to those with female headed households. Furthermore, the marginal effect reveals that a one year increase in the household head's age can possibly decrease the probability of being poor by 0.01 percentage points. As such, a decrease in the health expenditure as a proportion of total expenditure of the household by 1% will increase the probability of being poor by 0.19 percentage points.

Moreover, education level of the head of the household can also be indicated as a significant variable in defining and determining the prevalence of poverty. The findings denote that, for every increase in household head's educational level by one additional year, the probability of being poor will decrease by 0.50 percentage units. The estimated coefficients depict that, due to an increment in the proportion of employed household members in the workforce (age between 15 and 65) can decrease the likelihood of being in a poor household by 1.56 units.

When considering the marginal effect of the geographical location of household units, if the household is situated in the urban sector, probability of being poor will decrease by 1.22 percentage points in relation to households in rural and estate sectors.

The coefficients of different types of chronic illnesses disclose that the households suffer from high severe chronic conditions such as heart disease, blood pressure, cancer, kidney diseases, diabetes or asthma are significantly linked with poor households. When considering the marginal effects, being a household suffering from severe chronic diseases increases the likelihood of being a poor household by 1.39 percent units. With that said, households suffering from chronic conditions such as epilepsy, mental retardation or severe headache (brain diseases), have a higher probability of being a poor household by 5.01 percent units. Moreover, households suffering from eye diseases, ear diseases or catarrh (Ent_diseases), have a higher probability of being a poor household by 1.65 percent units. When considering the marginal effect of other types of chronic diseases such as arthritis, stomach diseases or haemorrhoids increases the likelihood of being a poor household by 1.90 percent units.

From the perspective of total household units, it was discovered that if a chronically ill person is deprived from access to basic education, lives in the rural or estate area, they become highly vulnerable to live in poverty because of chronic diseases. Further, probit regression

**Table 4. Probit model estimation results for household dataset, Sri Lanka.**

| Variable | Estimate | Robust SE | Marginal effect |
|---|---|---|---|
| Constant | -5.5081*** | 0.3007 | |
| **Socio-economic and demographic characteristics** | | | |
| Chro_ill_patients | -0.2545** | 0.1049 | -0.0142 |
| Males_HH | -0.2358*** | 0.0867 | -0.0137 |
| Elders_HH | 0.4850*** | 0.1273 | 0.0281 |
| Pr_hh_working | -0.2691*** | 0.0490 | -0.0156 |
| Male_headed | 0.0952* | 0.0536 | 0.0053 |
| Head_age | -0.0083*** | 0.0017 | -0.0005 |
| Maritalstatus_head | 0.1231** | 0.0559 | 0.0067 |
| Edu_level | -0.0867*** | 0.0045 | -0.0050 |
| **Ethnicity** | | | |
| Sinhala | 3.1586*** | 0.1979 | 0.1374 |
| Sri Lankan Tamil | 3.4641*** | 0.2501 | 0.8294 |
| Indian Tamil | 3.3206*** | 0.2599 | 0.8730 |
| Sri Lankan Moors | 3.3576*** | 0.2873 | 0.8531 |
| Malay | 3.7948*** | 0.4459 | 0.9415 |
| **Religion** | | | |
| Buddhist | 2.5392*** | 0.1765 | 0.1217 |
| Hindu | 2.4259*** | 0.1454 | 0.5263 |
| Islam | 2.3439*** | 0.2160 | 0.5536 |
| Roman Catholic/Other Christian | 2.3536*** | 0.1823 | 0.5693 |
| Health_exp | -3.4029*** | 0.6826 | -0.1971 |
| Head_chronic | -3.4029*** | 0.6826 | -0.1971 |
| **Geographical location** | | | |
| Urban | -0.2470*** | 0.0681 | -0.0122 |
| Colombo | -1.0571*** | 0.1472 | -0.0291 |
| Gampaha | -0.8875*** | 0.1264 | -0.0266 |
| Kalutara | -0.7511* | 0.1219 | -0.0234 |
| Kandy | -0.4586*** | 0.1087 | -0.0181 |
| Matale | -0.5810*** | 0.1301 | -0.0200 |
| Nuwara_Eliya | -0.6106*** | 0.1271 | -0.0208 |
| Galle | -0.6154*** | 0.1187 | -0.0214 |
| Matara | -0.2921*** | 0.1109 | -0.0131 |
| Hambantota | -0.9533*** | 0.1470 | -0.0249 |
| Jaffna | -0.3730*** | 0.0978 | -0.0154 |
| Mannar | -1.3527*** | 0.2389 | -0.0255 |
| Vavuniya | -1.2931*** | 0.2101 | -0.0254 |
| Batticaloa | -0.4363*** | 0.0953 | -0.0171 |
| Ampara | -0.8530*** | 0.1274 | -0.0239 |
| Trincomalee | -0.3839*** | 0.1117 | -0.0156 |
| Kurunegala | -0.6865*** | 0.1145 | -0.0231 |
| Puttalam | -0.7693*** | 0.1413 | -0.0229 |
| Anuradhapura | -0.5435*** | 0.1263 | -0.0195 |
| Polonnaruwa | -0.8560*** | 0.1560 | -0.0235 |
| Badulla | -0.1808 | 0.1141 | -0.0089 |
| Moneragala | -0.3603*** | 0.1238 | -0.0150 |
| Ratnapura | -0.3331*** | 0.1071 | -0.0144 |

*(Continued)*

**Table 4.** (Continued)

| Variable | Estimate | Robust SE | Marginal effect |
|---|---|---|---|
| Kegalle | -0.3087*** | 0.1129 | -0.0136 |
| **Type of chronic illness** | | | |
| High_sev_diseases | 0.2213** | 0.0997 | 0.0139 |
| Brain_diseases | 0.5346*** | 0.0959 | 0.0501 |
| Ent_diseases | 0.2308* | 0.1177 | 0.0165 |
| Otherdiseases | 0.2600** | 0.1038 | 0.0190 |
| Area under ROC curve | | | 0.7969 |
| Pseudo $R^2$ | | | 0.2437 |
| Log likelihood | | | -3230.4914 |
| No. of observation | | | 20,696 |

Source: Author's calculation based on the DCS [64].

Note

***significant at the 1% level

** significant at the 5% level; significant at the 10% level; Variable 'Burgher' was dropped with 26 observations because of estimability; Variables 'Buddhist', 'Kilinochchi' and 'Rural' were removed from the stepwise regression because p> = 0.15.

analysis identified that brain diseases such as, epilepsy, mental retardation and severe headache have a high propensity to push diseased victims towards poverty compared to other chronic diseases. Remarkably, results of this study revealed that high severe diseases such as, heart diseases, blood pressure, cancer, eye diseases, arthritis, stomach diseases and etc., have the probability of making victims less non-poor compared to other types of NCD's.

This study includes several limitations as well. Limitations of data in the HIES 2016, hindered researchers of this study from broadening the findings. In specific, the list of chronic diseases in the survey did not include several other major chronic diseases such as Multiple sclerosis, Parkinson disease and Crohn's disease, as defined by the U.S National Library of Medicine, and the disease of Obstructive pulmonary by the WHO. The omission of these risky diseases that are commonly known worldwide is a drawback; among the population of this study, there may be certain patients suffering from these diseases whose data has not been captured. Hence, significance of its harmful effects have not been incorporated to the study, when considering the impact of chronic diseases on the level of poverty.

## Conclusion and policy implications

It was discovered that in terms of total household units, patients suffering from brain diseases such as, epilepsy, mental retardation and severe headache are more vulnerable to fall into the trap of poverty compared to those with other chronic diseases included in the survey. Furthermore, if a chronically ill person is deprived from access to basic education and lives in rural or estate areas, they are more vulnerable to live in poverty.

It should be highlighted that heart disease, cancer, kidney diseases are increasing, and can be identified as the main diseases among NCDs in the Sri Lankan context. The chronic kidney disease (CKD) of unknown etiology is a killer among many farmers (head of the households) who cultivate rice—the staple food for Sri Lankans; CKD affects lives of family members of these farmer households in Sri Lanka leaving them with a feeling of insecurity, when they become physically unfit to engage in farming activities. This is also a potential area for future research, incorporating socio-economic issues, the study for a broader and pragmatic analysis. Policies can then be formulated in a more focused manner to enhance its viability.

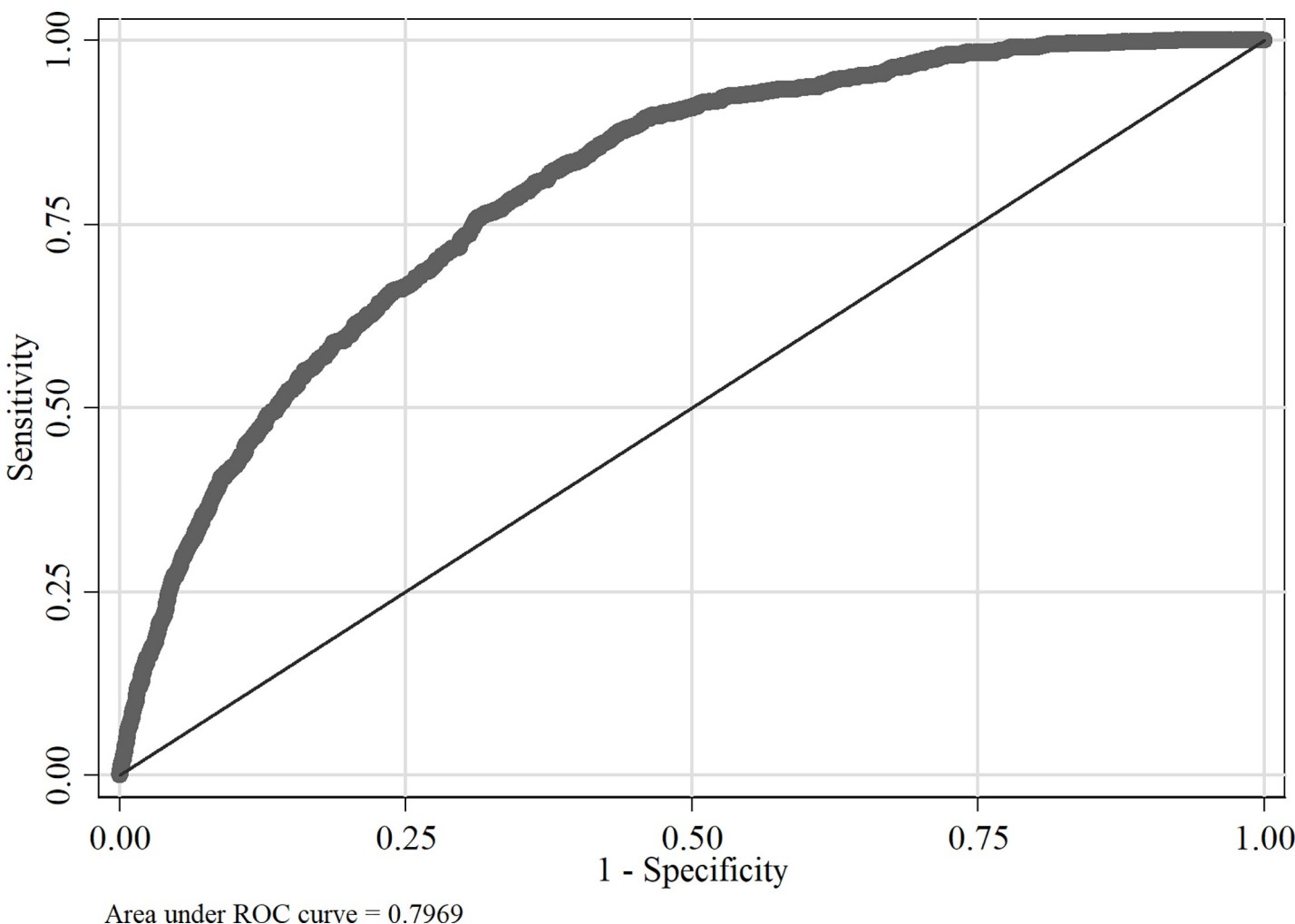

**Fig 2. ROC curve for household dataset.** Source: Author's illustration based on the DCS [64].

Most CKD patients are reported from rural areas, from low-income households and their employment is farming. Prevailing farming practices, fertiliser usage, water management etc., make these farmers vulnerable to chronic illnesses. In the case of CKD, it can be also assumed that these victims from rural area farming communities have received little primary education or some with no education at all. Therefore, lack of awareness of health related matters and the ability to make a living for a basic standard of life are rather limited. Thus, these contributory factors have pushed them into chronic illnesses and thereby to be entrapped in poverty. Hence, preventive action is the need of the hour. These include access to better education, awareness and regulation on safety practices for farmers, usage of wells for storage and consumption of water at the household level etc.

Another fact is that increased usage of alcohol, drugs, illicit liquor, smoking and tobacco, habits like chewing betel, sedentary lifestyles, increased consumption of fast food etc., have triggered the onset of cancer, heart diseases and blood pressure. This can be observed in both rural and urban areas, where the poor and non-poor, regardless of educational levels have become victims. Therefore, these issues can be addressed from the perspective of a preventive strategy to minimise the need for detecting illnesses and treatments. Such strategies and

policies can save high medical costs, thereby improving the quality of local health care facilities for the GOSL, specially in a backdrop where health care expenditure is faced with tight budgetary constraints. As for households, this can help reduce the non financial burden, stress and burnout associated with chronic illnesses. In other words, when a household member is suffering from chronic diseases, the burden in the long run has more implications on the family. Nutritional requirements, healthy food consumption habits, having an affordable yet a balanced diet and maintaining physical fitness can reap benefits for this purpose.

When women are employed and have received a better education level, it can prevent them and their households from chronic illnesses and thereby falling into poverty. Polices that empower women from the rural and estate sectors, and enabling them to be employed can uplift lives of this segment. Encouraging self-employment, initiating cottage industries, facilitating marketability for their produce and providing entrepreneurial guidance for those with low education levels, can make a noticeable impact on these women from low-income households. Such measures can enable women to support their families for out-of-pocket health care expenses.

Policy implications recommended via the study can be stated under the perspective of the GOSL to be considered for the decision making process. In accordance with the budget 2019 of the GOSL, 1.21% being allocated for healthcare facilities signals a gradual decrease in the proportion of government expenditure allotted on healthcare. Thus, an increment on government expenditure on stabilisation and development of healthcare facilities is considered to be an essential factor, to prevent or alleviate chronic illnesses [14] and to help in reducing the burden on health care expenditure. As such, having firm healthcare policies and regulated private healthcare sector can help solve issues associated with affordability [17].

Regulated private health care sector would also ease the affordability issue faced by the patients. Nevertheless, policy instruments such as creating public-private sector partnerships enable equitable access for healthcare facilities. Moreover, public and private sector collaborations strengthen private sector resources, resource sharing, sharing of expertise for an effective and a timely service offering. Few public-private interventions could be identified as feasible and popular such as contracting out, licensing, franchises and partnerships. The quality of healthcare facilities of private sector could be strengthened by extending licensing and accreditation systems to them. Such intervention methods have been successfully implemented in countries like Brazil, South Africa etc [67].

Furthermore, commercial insurance and community-based mutual services could be introduced with regard to healthcare facilities, specially for those suffering from brain diseases such as, epilepsy, mental retardation and severe headache. The developing countries like Colombia, Ghana and etc., have implemented such insurance schemes to reduce the financial burden, thus, endeavouring for affordable healthcare facilities with quality.

## Supporting information

**S1 Appendix. Characteristics of Sri Lankan households 2016.**
(DOCX)

**S2 Appendix. Initial probit model estimation results for household dataset, Sri Lanka.**
(DOCX)

## Acknowledgments

The authors would like to thank Dr. (Mrs.) I. R. Bandara, Director General of Department of Census and Statistics, Sri Lanka who granted permission to access data of the Household

Income and Expenditure Survey 2016. The authors also would like to thank Ms. Gayendri Karunarathne for proof-reading and editing this manuscript.

## Author Contributions

**Conceptualization:** Ruwan Jayathilaka, Sheron Joachim, Venuri Mallikarachchi, Nishali Perera, Dhanushika Ranawaka.

**Data curation:** Nishali Perera.

**Formal analysis:** Ruwan Jayathilaka, Sheron Joachim, Venuri Mallikarachchi, Nishali Perera, Dhanushika Ranawaka.

**Investigation:** Ruwan Jayathilaka.

**Methodology:** Ruwan Jayathilaka, Sheron Joachim, Venuri Mallikarachchi, Nishali Perera, Dhanushika Ranawaka.

**Supervision:** Ruwan Jayathilaka.

**Validation:** Ruwan Jayathilaka, Sheron Joachim.

**Visualization:** Venuri Mallikarachchi, Dhanushika Ranawaka.

**Writing – original draft:** Ruwan Jayathilaka, Sheron Joachim, Venuri Mallikarachchi, Nishali Perera, Dhanushika Ranawaka.

**Writing – review & editing:** Ruwan Jayathilaka.

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
