## [Decision Letter · Decision Letter 0]

19 Aug 2020

PONE-D-20-12031

Is there a link between chronic illness and the poverty?

PLOS ONE

Dear Dr. Jayathilaka,

Thank you for submitting your manuscript to PLOS ONE. After careful consideration, we feel that it has merit but does not fully meet PLOS ONE’s publication criteria as it currently stands. Therefore, we invite you to submit a revised version of the manuscript that addresses the points raised during the review process.

**Although I was unable to guarantee at least two Reviewers, despite inviting dozens of scholars, I believe I can make a decision on the basis of Reviewer 1's review.**

**I fully agree with the Reviewer 1 that, at the current state, the article is not suitable for publication. I invite the Authors to pay particular attention to point 1 of Reviewer 1's comments and to make a careful revision of study’s methodological aspects.**

We look forward to receiving your revised manuscript.

Kind regards,

Stefano Federici, Ph.D.

Academic Editor

PLOS ONE

Additional Editor Comments:

Although I was unable to guarantee at least two Reviewers, despite inviting dozens of scholars, I believe I can make a decision on the basis of Reviewer 1's review.

I fully agree with the Reviewer 1 that, at the current state, the article is not suitable for publication. I invite the Authors to pay particular attention to point 1 of Reviewer 1's comments and to make a careful revision of study’s methodological aspects.

3. Please edit your title to align with PLOS publication guidelines: https://journals.plos.org/plosone/s/submission-guidelines#loc-title In this case the title does not communicate the specific contentor scope of the research. Additionally there is a grammatical error, the word "the" should be removed.  Please amend both the title on the online submission form (via Edit Submission) and in the manuscript so that they are identical.

5. Please review your table titles. Please include a copy of Table 4 which you refer to in your text on page 11 and Table 5 which you refer to in your text on pages 12 and 13.

Reviewers' comments:

Reviewer's Responses to Questions

**Comments to the Author**

1. Is the manuscript technically sound, and do the data support the conclusions?

Reviewer #1: Partly

2. Has the statistical analysis been performed appropriately and rigorously? 

Reviewer #1: No

3. Have the authors made all data underlying the findings in their manuscript fully available?

Reviewer #1: Yes

4. Is the manuscript presented in an intelligible fashion and written in standard English?

Reviewer #1: Yes

5. Review Comments to the Author

Reviewer #1: The manuscript titled "Is there a link between chronic illness and the poverty" looks whether chronic illnesses affect the poverty status of a household in Sri Lanka. The authors construct a probit regression model where the dependent variable is a dummy variable - where 0 indicates the household is not in poverty, and 1 otherwise. The authors find that a household with chronically ill person is more likely to be in poverty.

Major Concerns:

1. The authors need to focus their literature review on literature concerning developing countries. The part where they talk about how expenditure on chronic illness affect poverty is appropriate. However, it is difficult to grasp why low educational attainment and marital status affect chronic illness. The authors should not make such a statement, and instead should offer an explanation how they could cause chronic illness (eg, low education means individual may have to do strenuous labor or work in hazardous conditions to earn a living, which increases their chances of getting a chronic illness).

2. The authors should also talk about the types of chronic illnesses more prevalent in developing countries. Are they bacterial (like TB), or maybe genetic (eg diabetes)?

3. The policy implications has to be tailored according to the type of chronic illnesses prevalent in Sri Lanka.

4. Per capita income must not be used as an independent variable in this case, as it and poverty status are highly correlated. Instead, the authors can use healthcare expenditure of household (which is available in HIES) as an independent variable. Or, the proportion of total household expenditure devoted to healthcare.

5. The authors do not include proportion elders in household, although it is included in the summary statistics. The authors should include that variable in the probit regression.

6. The effect of chronic illness on poverty should be dependent on whether the head of household as it or not. I would suggest adding interaction terms of head of household with each of the four chronic illnesses in the model. They will then show if the poverty status of household is even more affected if the head of household has that illness.

7. I am not sure if 'average age of household' is an appropriate variable. Maybe a better independent variable would be 'age of household head' or 'age of oldest member of household'

8. There is some issues with the observation count. The HIES surveyed 25,640 households, but the regression only has 21,756 household. Why this difference?

9. Ethnicity and religion are presented as dummy variables. What are they measuring? Is ethnicity=1 Tamil? Singhalese? Is religion=1 Hindu? Buddhist? Muslim?

10. Instead of labourforce_HH variable, the authors could look at proportion of household members working. employed/(labourforce in HH).

11. The tables are not properly numbered.

12. A few more regressions should be run to see if the significance are robust.

13. Since the paper is about the link of chronic illness on poverty, the authors should discuss more about how chronic illness affect poverty in the "results and discussion" section.

14. The authors should also consider including district fixed effects in the regression.

6. PLOS authors have the option to publish the peer review history of their article (what does this mean?). If published, this will include your full peer review and any attached files.

Reviewer #1: No

---

## [Author Response · Author response to Decision Letter 0]

1 Oct 2020

Reviewer 1: 

The manuscript titled "Is there a link between chronic illness and the poverty" looks whether chronic illnesses affect the poverty status of a household in Sri Lanka. The authors construct a probit regression model where the dependent variable is a dummy variable - where 0 indicates the household is not in poverty, and 1 otherwise. The authors find that a household with chronically ill person is more likely to be in poverty.

Comments to Reviewer: 

The manuscript title has been slightly modified as follows to express the impact of chronic illness on poverty.

“Do Chronic Illnesses and Poverty Go Hand in Hand?”

Reviewer 1: Major Concerns:

1. The authors need to focus their literature review on literature concerning developing countries. The part where they talk about how expenditure on chronic illness affect poverty is appropriate. However, it is difficult to grasp why low educational attainment and marital status affect chronic illness. The authors should not make such a statement, and instead should offer an explanation how they could cause chronic illness (eg, low education means individual may have to do strenuous labor or work in hazardous conditions to earn a living, which increases their chances of getting a chronic illness).

Comments to Reviewer: 

The following new paragraph has been added to the literature review section to include developing countries in the literature review.

“In a study based in Sri Lanka, Pallegedara (42) examined the effects of chronic NCDs on out-of-pocket health expenditures of households. Findings revealed that medical poverty is high among chronic NCDs. In another study, Pallegedara and Grimm (43) stressed that older persons are more inclined to suffer from chronic diseases. The two-part model based on the 2012/2013 household survey was carried out to examine the association of NCD-prevalence and healthcare utilization with household consumption [44]. They found that the relationship between private healthcare utilization and household consumption was negative. In another study, Kumara and Samaratunge (45) investigated the patterns and determinants of the burden of expenses in households. Findings confirmed that the burden of expenses does not vary significantly with the variation in income.”

The following sentence has been deleted 

“Therefore, most findings suggest that married individuals are more exposed to chronic conditions and tend to incur high medication costs.”

The following sentence has been included.

“On the other hand, low education means individual may have to do jobs that involve strenuous labour or work in hazardous conditions to earn a living, which increases their chances of getting a chronic illness.”

Reviewer 1: 

2. The authors should also talk about the types of chronic illnesses more prevalent in developing countries. Are they bacterial (like TB), or maybe genetic (eg diabetes)?

Comments to Reviewer: 

A couple of sentences have been added to emphasize the types of chronic illness in the developing countries. Final paragraph has been re-written as follows,

“Chronic illnesses such as diabetes mellitus, cardiovascular disease, cerebrovascular disease, cancers, chronic respiratory diseases, mental illnesses, and other NCDs have become the major cause of death and disability of citizens in countries worldwide[6], where more than two third of all deaths are caused by a certain type of a chronic disease [7]. More than 41 million people have died from NCDs in 2016, out of which 15 million of these deaths have occurred between the age of 30-70 years, mainly from cardiovascular diseases (31%), cancers (16%), chronic respiratory diseases (7%), diabetes (3%) and other NCDs (15%) [3]. According to WHO estimates, nearly 78% of total deaths caused by chronic diseases have occurred in LMICs in 2016 [8]. This is due to mainly four types of NCDs – cardiovascular diseases, cancers, chronic respiratory diseases and diabetes. On average, NCDs are projected to be the major cause of disease burden worldwide, exceeding communicable diseases, puerperal, prenatal, and food diseases in every country; chronic diseases are recognised as the origin of disability as 68% of people living with disability worldwide, and 84% of people living in LMICs [9]. If current growth trends persist by 2020, 7 out of every 10 deaths in developing countries will be attributed to NCDs and NCDs deaths worldwide by 2030 will be 52 million [10]. A research study in the USA states that, 12% of the child population is likely to suffer from a certain type of a NCD, with asthma being the most common illness next to allergy and obesity; this is predicted to rise in the coming years [11]. Thus, the increasing prevalence of multiple chronic conditions will adversely affect every country across the globe and result in high healthcare utilisation and increasing costs.”

Reviewer 1: 

3. The policy implications has to be tailored according to the type of chronic illnesses prevalent in Sri Lanka.

Comments to Reviewer: 

This section has now been revised. The following four paragraphs have been added to the manuscript.

“It should be highlighted that heart disease, cancer, kidney diseases are increasing, and can be identified as the main diseases among NCDs in the Sri Lankan context. The chronic kidney disease (CKD) of unknown etiology is a killer among many farmers (head of the households) who cultivate rice - the staple food for Sri Lankans; CKD affects lives of family members of these farmer households in Sri Lanka leaving them with a feeling of insecurity, when they become physically unfit to engage in farming activities. This is also a potential area for future research, incorporating socio-economic issues, the study for a broader and pragmatic analysis. Policies can then be formulated in a more focused manner to enhance its viability.

Most CKD patients are reported from rural areas, from low-income households and their employment is farming. Prevailing farming practices, fertiliser usage, water management etc., make these farmers vulnerable to chronic illnesses. In the case of CKD, it can be also assumed that these victims from rural area farming communities have received little primary education or some with no education at all. Therefore, lack of awareness of health related matters and the ability to make a living for a basic standard of life are rather limited. Thus, these contributory factors have pushed them into chronic illnesses and thereby to be entrapped in poverty. Hence, preventive action is the need of the hour. These include access to better education, awareness and regulation on safety practices for farmers, usage of wells for storage and consumption of water at the household level etc.

Another fact is that increased usage of alcohol, drugs, illicit liquor, smoking and tobacco, habits like chewing betel, sedentary lifestyles, increased consumption of fast food etc., have triggered the onset of cancer, heart diseases and blood pressure. This can be observed in both rural and urban areas, where the poor and non-poor, regardless of educational levels have become victims. Therefore, these issues can be addressed from the perspective of a preventive strategy to minimise the need for detecting illnesses and treatments. Such strategies and policies can save high medical costs, thereby improving the quality of local health care facilities for the GOSL, specially in a backdrop where health care expenditure is faced with tight budgetary constraints. As for households, this can help reduce the non financial burden, stress and burnout associated with chronic illnesses. In other words, when a household member is suffering from chronic diseases, the burden in the long run has more implications on the family. Nutritional requirements, healthy food consumption habits, having an affordable yet a balanced diet and maintaining physical fitness can reap benefits for this purpose.

When women are employed and have received a better education level, it can prevent them and their households from chronic illnesses and thereby falling into poverty. Polices that empower women from the rural and estate sectors, and enabling them to be employed can uplift lives of this segment. Encouraging self-employment, initiating cottage industries, facilitating marketability for their produce and providing entrepreneurial guidance for those with low education levels, can make a noticeable impact on these women from low-income households. Such measures can enable women to support their families for out-of-pocket health care expenses.”

Reviewer 1: 

4. Per capita income must not be used as an independent variable in this case, as it and poverty status are highly correlated. Instead, the authors can use healthcare expenditure of household (which is available in HIES) as an independent variable. Or, the proportion of total household expenditure devoted to healthcare.

Comments to Reviewer: 

As suggested, a new variable “Health expenditure as a proportion of total expenditure (Health_exp)” has been added to the model and “income Decile (Per_income_level)” has been deleted.

The new variable is statistically significant. Thus, the new variable has been kept in the model and the interpretation and text has been adjusted accordingly.

Reviewer 1: 

5. The authors do not include proportion elders in household, although it is included in the summary statistics. The authors should include that variable in the probit regression.

Comments to Reviewer: This was a result of the stepwise technique. New variables were selected with p-value <0.10 and previously selected variables removed with p-value ≥0.15. So the final model did not include the variable “proportion elders in household”. However, after adding a few suggested variables, the stepwise model dropped ‘Burgher’ variable with 26 observation. Moreover, the variables ‘Buddhist’, ‘Kilinochchi’ and ‘Rural’ were also removed from the final stepwise regression because p>=0.15. To make this procedure clear, further explanations have been added to the methodology section and the results section. For instance, the following sentences have been added to the note under regression results table.

“Variable ‘Burgher’ was dropped with 26 observations because of estimability; Variables ‘Buddhist’, ‘Kilinochchi’ and ‘Rural’ were removed from the stepwise regression because p>=0.15.”

Reviewer 1: 

6. The effect of chronic illness on poverty should be dependent on whether the head of household as it or not. I would suggest adding interaction terms of head of household with each of the four chronic illnesses in the model. They will then show if the poverty status of household is even more affected if the head of household has that illness.

Comments to Reviewer: 

As suggested, household head is a chronic patient (Head_chronic) information has been added as an interaction term to the model. The newly created variable is significant. The interpretation in the text has been changed accordingly.

Reviewer 1: 

7. I am not sure if 'average age of household' is an appropriate variable. Maybe a better independent variable would be 'age of household head' or 'age of oldest member of household'

Comments to Reviewer: 

As suggested, Head_age (Age of the household head (in years)) has been added to the model. Since the added variable is significant and more meaningful the old variable (Avg_age) has been, dropped. Interpretation and text has been changed accordingly.

Reviewer 1: 

8. There is some issues with the observation count. The HIES surveyed 25,640 households, but the regression only has 21,756 household. Why this difference?

Comments to Reviewer: 

Comment is noted. The HIES consists of 25,640 housing units. However, only 21,756 households responded. The sentence has been corrected as follows.

“The sample consists of 25,640 housing units which includes 21,756 responded households, covering all 25 districts in the country and aims to analyse the seasonal and regional differences of income and expenditure levels together with buying patterns of households.”

Reviewer 1: 

9. Ethnicity and religion are presented as dummy variables. What are they measuring? Is ethnicity=1 Tamil? Singhalese? Is religion=1 Hindu? Buddhist? Muslim?

Comments to Reviewer: 

A few dummies were added to represent the Ethnicity and religion as control variables. For instance, separate dummy variables for ‘Sinhala’, ‘Sri Lankan Tamil’, ‘Indian Tamil’, ‘Sri Lankan Moors’, ‘Malay’, ‘Burgher’, and ‘Other’; ‘Other’ has been considered as the reference category. Likewise, separate dummy variables for Buddhist, Hindu, Islam, Roman Catholic/Other Christian, and Other; Other has been considered as the reference category.

Since the study has used several dummy variables, the characteristics of Sri Lankan households table has been expanded. Thus, it has been moved to the Appendix A1 in the revised version.

Reviewer 1: 

10. Instead of labourforce_HH variable, the authors could look at proportion of household members working. employed/(labourforce in HH).

Comments to Reviewer: 

Labourforce_HH variable was dropped from the model. As suggested a new variable was created as ‘proportion of household members working’; (employed household members/Labour force in the household). Newly created variable name is Pr_hh_working. New variable is significant and has been kept in the model. In addition, interpretation has been changed accordingly.

Reviewer 1: 

11. The tables are not properly numbered.

Comments to Reviewer: 

Noted and Thank you. This has been corrected in the revised version. 

Reviewer 1: 

12. A few more regressions should be run to see if the significance are robust.

Comments to Reviewer: 

Full regression was estimated before finalising the stepwise probit model. Appendix A2 shows the results of the initial regression. 

Reviewer 1: 

13. Since the paper is about the link of chronic illness on poverty, the authors should discuss more about how chronic illness affect poverty in the "results and discussion" section.

Comments to Reviewer: 

This has been corrected in the revised version.

Reviewer 1: 

14. The authors should also consider including district fixed effects in the regression.

Comments to Reviewer: 

This has been corrected in the revised version. 

For instance, separate dummy variables for ‘Colombo’, ‘Gampaha’, ‘Kalutara’, ‘Kandy’, ‘Matale’, ‘Nuwara_Eliya’, ‘Galle’, ‘Matara’, ‘Hambantota’, ‘Jaffna’, ‘Mannar’, ‘Vavuniya’, ‘Kilinochchi’, ‘Batticaloa’, ‘Ampara’, ‘Trincomalee’, ‘Kurunegala’, ‘Puttalam’, ‘Anuradhapura’, ‘Polonnaruwa’, ‘Badulla’, ‘Moneragala’, ‘Ratnapura’, and ‘Kegalle’ districts used; ‘Mullaitivu’ has been considered as the reference category.

---

## [Decision Letter · Decision Letter 1]

12 Oct 2020

Do chronic illnesses and poverty go hand in hand?

PONE-D-20-12031R1

Dear Dr. Jayathilaka,

We’re pleased to inform you that your manuscript has been judged scientifically suitable for publication and will be formally accepted for publication once it meets all outstanding technical requirements.

Kind regards,

Stefano Federici, Ph.D.

Academic Editor

PLOS ONE

Additional Editor Comments (optional):

Reviewers' comments:

Reviewer's Responses to Questions

**Comments to the Author**

1. If the authors have adequately addressed your comments raised in a previous round of review and you feel that this manuscript is now acceptable for publication, you may indicate that here to bypass the “Comments to the Author” section, enter your conflict of interest statement in the “Confidential to Editor” section, and submit your "Accept" recommendation.

Reviewer #1: All comments have been addressed

2. Is the manuscript technically sound, and do the data support the conclusions?

Reviewer #1: Yes

3. Has the statistical analysis been performed appropriately and rigorously? 

Reviewer #1: Yes

4. Have the authors made all data underlying the findings in their manuscript fully available?

Reviewer #1: (No Response)

5. Is the manuscript presented in an intelligible fashion and written in standard English?

Reviewer #1: Yes

6. Review Comments to the Author

Reviewer #1: The authors have addressed all my concerns. I am happy to say that the manuscript is ready for publication in PLOS ONE.

7. PLOS authors have the option to publish the peer review history of their article (what does this mean?). If published, this will include your full peer review and any attached files.

Reviewer #1: No

---

## [Editor Report · Acceptance letter]

15 Oct 2020

PONE-D-20-12031R1 

Do chronic illnesses and poverty go hand in hand? 

Dear Dr. Jayathilaka:

I'm pleased to inform you that your manuscript has been deemed suitable for publication in PLOS ONE. Congratulations! Your manuscript is now with our production department. 

Kind regards, 

on behalf of

Prof. Stefano Federici 

Academic Editor

PLOS ONE